# Plasticity in Chemical Host Plant Recognition in Herbivorous Insects and Its Implication for Pest Control

**DOI:** 10.3390/biology11121842

**Published:** 2022-12-16

**Authors:** Sylvia Anton, Anne-Marie Cortesero

**Affiliations:** 1IGEPP, INRAE, Institut Agro, Univ Rennes, CEDEX, 49045 Angers, France; 2IGEPP, INRAE, Institut Agro, Univ Rennes, CEDEX, 35042 Rennes, France

**Keywords:** insect herbivore, behavior, pest control, chemical host plant cue, host recognition, long- and short-range recognition, experience, physiological state

## Abstract

**Simple Summary:**

Insects feeding on plants constitute a serious problem for agriculture because they produce large populations, especially in monocultures. Insect pests are mainly controlled with chemical pesticides, but with major problems caused by the development of resistance and toxic effects on non-target organisms, alternative management of insect pests has become an important goal. Understanding how insects find and choose their host plants is one major research line in order to develop environmentally acceptable methods to protect cropped plants. We review here information on how herbivorous insects use chemical cues, such as volatiles and contact cues, in a sequential way to find and finally choose their host plants. Then we put forward evidence that insect behavior, guided by chemical cues, strongly depends on the chemical environment and in addition varies as a function of the insect’s physiological state, i.e., if they are sexually mature, have mated, have fed, and had previous experience, thus displaying a high degree of plasticity. In order to successfully manipulate pest insect behavior for sustainable crop protection, we need to implicate and further increase our knowledge firstly on interactions between chemical cues and their environment, and secondly on the different types of plasticity.

**Abstract:**

Chemical communication is very important in herbivorous insects, with many species being important agricultural pests. They often use olfactory cues to find their host plants at a distance and evaluate their suitability upon contact with non-volatile cues. Responses to such cues are modulated through interactions between various stimuli of biotic and abiotic origin. In addition, the response to the same stimulus can vary as a function of, for example, previous experience, age, mating state, sex, and morph. Here we summarize recent advances in the understanding of plant localization and recognition in herbivorous insects with a focus on the interplay between long- and short-range signals in a complex environment. We then describe recent findings illustrating different types of plasticity in insect plant choice behavior and the underlying neuronal mechanisms at different levels of the chemosensory pathway. In the context of strong efforts to replace synthetic insecticides with alternative pest control methods, understanding combined effects between long- and close-range chemical cues in herbivore–plant interactions and their complex environment in host choice are crucial to develop effective plant protection methods. Furthermore, plasticity of behavioral and neuronal responses to chemical cues needs to be taken into account to develop effective sustainable pest insect control through behavioral manipulation.

## 1. Introduction

Herbivorous insects have a significant impact on crop production and many species are considered as serious pests. Insect pest control relies mainly on insecticides, but their toxicity towards “beneficial” insects and other organisms, including humans, as well as their strong general impact on the environment, makes the search for alternative control methods a major challenge. As localizing and choosing a host plant is a crucial step for herbivorous insects to colonize plants, interfering with these behaviors is considered a powerful strategy to protect crops in the future. Host choice is an important issue for all insects feeding on plants in order to identify optimal food. For female herbivorous insects looking for a place to lay eggs, this is even more important, in order to provide the optimal substrate for the development of their offspring. Host plant choice is nevertheless not only performed by adult females, but larvae can also approach or switch host plants, even if often only over shorter distances. Males are also attracted by host plants, either for feeding or for localization of an environment, which likely houses a mating partner. Cues provided by host plants that can potentially be used depend on the degree of specialization (mono-, oligo-, or polyphagous), on the environment, and on the mobility of the concerned insect. Even though other sensory modalities can play a role in host plant localization and choice, chemical cues are of utmost importance in a large majority of cases. Generally, different cues are used sequentially at different distances from the host plant, up to surface contact and finally probing to evaluate suitability. Insect responses to these cues vary, however, as function of environmental factors, previous experience, and physiological state. Therefore, these factors of plasticity need to be considered when trying to modify herbivorous insect behavior for plant protection.

Whereas plasticity and modulation of insect olfactory-guided behavior in general and the underlying neuronal mechanisms have been reviewed recently [1], we will in this review summarize the recent literature on what is known to date on how insects use chemical long- and short-range cues to choose their host plants and how different environmental cues may influence sensory systems and orientation behavior. Different forms of plasticity in host plant choice, caused by environmental factors, experience, and the physiological state of an insect will be illustrated by examples from the recent literature and their role in adaptation to host plants will be discussed. We will not cover here the evolutionary arms race concerning chemical signals between plants and insects, such as the production of toxic plant compounds and adaptation of insect metabolisms, as reviewed in [2]. Finally, we will put this knowledge into the context of plant protection against insect herbivores and discuss the potential use of the gained knowledge in the development of alternative control methods. 

## 2. Sequential Chemical Host Seeking Cues: Multiple Steps Involving Olfaction and Taste

It has been shown previously that insects use multiple cues in a sequential way to orient towards a host plant in a natural environment. While each case is different in detail, there are certain common principles which have been reviewed extensively [3]. Sequential host seeking steps have been described in various insects. Long-range detection of host plants often depends on the volatile organic compounds (VOCs) emitted by these plants. However, different strategies in host seeking at a distance must be distinguished. Many flying insects can actively orient towards attractive volatile cues, whereas others, such as aphids and other very small insects or larger species with poor flying abilities, are passively transported by air movements, or simply do not move over larger distances and make their decisions on host plant acceptance only at short distances, potentially involving both volatile and contact cues. The situation is still different in soil insect herbivores, i.e., root-feeding insects, which also use VOCs to orient towards a host plant at a (small) distance when walking/digging in the ground (reviewed in [4]).

Actively orienting insects can rely on different volatile cues, with non-specific habitat cues used first as an orientation guide and more specific volatiles are used in the second step for the final host choice [5]. This is especially true for generalist herbivores, which respond to and are attracted by a large variety of volatiles, and consequently use more specific volatiles of a more preferred host [6]. Certain plants emit highly specific compounds, which are often signaling non-palatability or even toxicity. However, insects that have adapted their metabolism to feed on these plants are attracted by these specific compounds. For example, in the cabbage fly *Delia radicum*, allyl-iso-thiocyanate acts as an attractant [7] and the onion fly, *Delia antiqua*, is strongly attracted by dipropyl disulfide [8]. In these cases, individual compounds can have a strong behavioral effect, but it is more common that only specific blends of compounds elicit attraction to host plants [9]. This is especially true for common VOCs emitted by a large variety of plants, but in different ratios depending on the plant species. Prominent examples for the importance of ratios of compounds in host plant volatile mixture recognition are tortricid moths, in which both behavioral responses to host plant volatiles and underlying neuronal mechanisms have been investigated. For example, females of the grapevine moth *Lobesia botrana* were shown to be attracted by specific ratios of ubiquitous volatiles corresponding to the ones emitted by the grapevine *Vitis vinifera* [10]. In this species rather ubiquitous compounds, such as E-2-hexenal, elicited responses in a large proportion of olfactory projection neurons within the primary olfactory center, the antennal lobe. While some neurons responded specifically to this compound, more than 50% of all investigated neurons responded to a large variety of compounds, thus possibly representing neurons involved in the processing of behaviorally active blends [11]. The ratio of compounds in host plant mixtures was also important in the oriental fruit moth, *Grapholita molesta*, a pest of peaches and other fruit. Even though varying the amount of one compound of peach shoot volatiles in a mixture allowed for some flexibility in the behavioral response, larger changes inhibited attraction in a Y-olfactometer. Monitoring activation of antennal lobe glomeruli responding to the mixture used for stimulation by calcium imaging, revealed that one glomerulus was highly sensitive to changes in the compound ratios, whereas another glomerulus responded similarly to a wide array of blends [12]. 

Concerning small insects with limited flying capacities, the use of sequential host cues has been studied extensively in a few aphid species. Long-range cues in aphids can be visual: colors, contrast between target and background, as well as visual patterns and shapes can be involved in guiding aphids to their hosts. However, similar to many other insects, some aphid species also use individual plant volatiles or specific blends of compounds to localize host plants and avoid non-host plants, but generally at shorter distances than actively flying insects (reviewed in [13]). In other poor flyers, such as the tortoise beetle *Cassida canaliculata*, host plant odors only cause a weak attraction and strong preferences are only elicited when contact with the plant surface is established [14].

For both types of herbivorous insects, different types of contact cues are required to provide additional information on the identity and quality of a host plant in order to make a final choice and to accept to oviposit or feed on it. For acceptance, the balance of phagostimulants and antifeedants is crucial, whereas a decision to feed on a plant after tasting it depends on a trade-off between the presence of toxic compounds and the nutritional value. In any case, even at short range, sequential cues are employed for host choice and final acceptance. In the caterpillar of the silk moth, *Bombyx mori*, for example, specific blends of leaf-surface compounds allow host plant recognition, but final acceptance depends on sugar detection in the leaf sap after biting [15]. 

Our understanding of behavioral host choice through contact chemical stimuli, has been improved through the investigation of gustatory receptors (GRs) and gustatory receptor neuron function. In several insects, the specific role of GRs in host recognition and acceptance has been identified. In *B. mori*, highly sensitive coumarin receptor neurons expressing the gustatory receptors BmGr53 and BmGr19 are present on the larval maxillary palp and detect minute amounts of this toxic compound at the leaf surface of non-host plants, such as the cherry plant, and suppress test biting, thus avoiding ingestion of highly harmful compounds [16]. In swallowtail butterflies, which are highly specialized on one or a few host plants, the presence of specific plant surface compounds is needed for females to accept a host plant for laying eggs. This is also the case in *Papilio xuthus*, where a highly specific gustatory receptor PxutGr1 was shown to be responsible for the detection of the surface compound synephrine, eliciting egg-laying behavior [17]. 

In pierid butterfly larvae, which are adapted to glucosinolates present in Brassicaceae plants, non-volatile molecules, such as glucomoringin for example, stimulate feeding. However, glucosinolates are precursors for toxic isothiocyanates, which can be present at the feeding site in dissolved form but can also enter the gas phase and be detected by olfactory receptors. Glucomoringin is a precursor for the non-volatile isothiocyanate moringin, known to be toxic for insect larvae. This compound elicited deterrent behavior at high doses and was shown to be detected by deterrent-sensitive gustatory neurons on the maxillary palps [18]. In the larvae of silk moths, *B. mori*, several putative bitter GRs were molecularly characterized and responded widely to different feeding deterrents, indicating that most bitter GRs might play a role in host plant recognition [19]. A much more specific and unexpected role of ionotropic and gustatory receptors has been described in *Drosophila melanogaster*. Whereas sourness is known to be a deterrent in most insects, lactic acid is a feeding stimulant in this species. The onset of feeding behavior upon lactic acid contact is mediated by a highly specific ionotropic receptor (IR25a), but parallel activation of sweet gustatory receptors (Gr64a-f) is necessary to elicit activity in a specific set of sensory neurons, leading to feeding behavior [20].

## 3. Chemical Cue Interaction Leads to Plasticity in Host Plant Recognition

Olfactory cues indicating host plants, similar to cues used for intraspecific communication, do not occur alone in a natural environment but within a background of multiple odors emitted from biotic and abiotic sources [21]. Concerning plant volatile emissions, different mechanisms of interaction for host- and non-host volatiles have been discovered (Table 1). Non-host volatiles can mask host volatiles without being repellent themselves or alternatively have a repulsive effect, which can reduce or override the attractive effect of host plant volatiles. Thus, insects must integrate olfactory information on attractive and repulsive volatiles and weigh opposing inputs to take an adaptive behavioral decision. Non-host plant volatiles can also indirectly affect host plant recognition by influencing volatile production in the host plant.

Odor-masking of attractive host plant volatiles in complex environments has been discussed previously, but actual examples of masking effects without repellency of the associated non-host volatiles are rather scarce [22,23]. It has been shown, however, in mixed cropping systems, where attraction of the Colorado potato beetle, *Leptinotarsa decemlineata*, to potato volatiles was reduced in an environment of wild tomato and cabbage plants, while the associated plants were not repellent by themselves (e.g., [24]). Odor-masking was also suggested in the case of the whitefly, *Bemisia tabaci*, in intercropping systems of tomato and coriander; while coriander alone was not repulsive for whiteflies and tomato plants were highly attractive, volatiles emitted by both plants together strongly reduced the attractiveness compared to tomato alone [25]. In most other cases, where authors report masking effects of non-host plant odors, this is combined with a repellent effect of non-host plants ([22,23] and references therein). Bark beetles, *Scolitinae*, represent another example for integration of contradicting olfactory cues used for host plant identification. Both attractive insect-produced and plant-emitted odors are integrated in combination with repulsive non-host cues to identify suitable hosts [26]. However, the neural mechanisms of compound interactions are not well understood. In these species, it is only known that antennal receptor neurons detecting attractive or repellent compounds seem to be clearly distinct, which indicates that compound interactions most likely happen within the central nervous system ([26] and references therein) (Figure 1).

**Table 1 biology-11-01842-t001:** Literature overview over different origins of chemosensory plasticity in host plant recognition of herbivorous insects.

Origin of Plasticity	Insect Taxa	References
** *Plasticity through plant cue interaction* **		
Interactions between chemical cues	Beetles	[24,26,27]
	Whiteflies	[25]
	Psyllids	[28]
	Aphids	[29,30,31,32]
Interactions between visual and chemical cues	Beetles	[33]
	Flies	[34,35]
	Psyllids	[36,37]
	Moths	[39,40]
** *Insect-dependent plasticity* **		
Chemosensory experience	Moth/butterflies	[41,42,43,44,45,46,47,48,55,59,60,61,62]
	Beetles	[49,51,56]
	Psyllids	[52]
	Locusts	[53,54,63]
	Cockroaches	[64]
Physiological state	Flies	[65,68,69,70,71,76,77,80,81]
	Aphids	[72]
	Psyllids	[37,78]
	Moths	[79]
Sex and morph	Moth	[82,83,84,85,86,87]
	Beetles	[88]
	Flies	[89]
	Aphids	[30,90]
	Locusts	[94,95,96,97,98,99]

A specific type of odor interaction occurs when general plant-emitted volatiles are behaviorally active as individual compounds. Recognition might in this case rather be mediated by a lack of additional compounds otherwise present in combination with the attractive cue in non-host plants. For the chrysomelid beetle *Galerucella vittaticollis*, a single ubiquitous VOC, cis-3-hexenyl acetate, emitted in large amounts by their preferred host plant, strawberry, was shown to be highly attractive. This volatile is, however, also emitted by many non-host plants. Thus, it seems that repellent non-host plants are recognized by specific ratios and combinations of cis-3-hexenly acetate with other volatiles [27].

An influence of non-host plant volatiles on the emission of volatiles by host plants has been observed in some cases. An example for such interactions has been discovered in host plants of the Asian citrus psyllid *D. citri*. Citrus host plants exposed to guava plants or to guava shoot odors for at least two hours were less attractive to the citrus psyllid than un-exposed plants [28]. Several studies have also examined the effect of companion plants on the attraction of aphids to their host plants. Certain weeds, such as *Chenopodium album*, reduce the acceptance of barley by *Rhopalosiphum padi* [29]. Potato plants exposed to volatiles from onion plants change their volatile emission profile subsequently, with increased amounts of two terpenoid compounds. As a consequence, *Myzus persicae* aphids were less attracted by onion-exposed potato plants than by unexposed plants [30,31]. Volatiles of flowering Marigold (*Tagetes* spp.) also decrease the attractivity of pepper plants (*Capsicum annuum*) for the aphid *M. persicae* [32].

## 4. Chemical Host Plant Recognition Is Modulated by Visual Cues 

In many insect species, chemical cues, especially volatiles, seem to be sufficient to localize and identify a suitable host plant. In other insects, attraction to a host plant is strongly enhanced when visual cues are added to olfactory cues (Table 1). For example, visual and olfactory cues act synergistically in ambrosia beetles during localization of their conifer host trees [33]. Synergistic attraction of visual cues (i.e., shapes and colors) combined with host plant odors have also been found in female melon flies, *Bactrocera cucurbitae* [34]. Highly plastic interactions between colors and volatiles playing a role in the attraction of *Drosophila suzukii* have also been found recently. Depending on the volatiles or volatile blends present, females may choose preferentially different colors, but synergistic effects between odors and colors depend on the origin of the presented odors [35]. In the Asian citrus psyllid, *D. citri*, a combination of citrus odors with yellow-colored paper positioned in the corresponding olfactometer fields increased attractivity [36]. In the same species, visual cues, i.e., a green LED to simulate a green plant, enhanced attraction to volatile cues [37]. Interactions and synergies between chemical and visual stimuli are often dependent on ecological factors [38] but may also vary as a function of mating status (see Section 6.2 for psyllids for example). The neurobiology of interactions between olfactory and visual cues has been investigated mainly in flower-visiting, pollinating insects, such as moths [38]. Interactions at different levels of the olfactory pathway have been shown in the moth *Manduca sexta*. Whereas responses to olfactory stimuli (flower odors) were not influenced by visual cues at the level of the antennal lobe, such interactions occurred within higher order integration centers, the mushroom bodies, but the type of effect (synergy or suppression) was odor-dependent [39,40].

## 5. Experience-Dependent Plasticity in Host Plant Responses

### 5.1. Plasticity Due to Olfactory Experience

Experience has been suggested to modulate host plant choice in various ways, especially in generalist species (Table 1). Larval odor experience during feeding can influence adult host choice, but adult experience during hatching, early adult life, successful mating on a host plant, and egg-laying may also modify subsequent host choice in insects. Several studies described these phenomena at the behavioral level and investigated underlying mechanisms. In the noctuid moth *Spodoptera littoralis* for example, both larval experience during feeding at different developmental stages on a plant, early adult experience, and mating on a specific host plant influence the preferences in the choice of an oviposition site in males and females [41,42,43,44]. 

Odor learning in larvae is often only possible during a specific sensitive period. In *S. littoralis* for example, host plant preference is modified by experience during late larval stages, but not when early larval stages feed on a plant and are subsequently fed on an artificial diet [45]. The hypothesis that experience might also influence behavioral choice in the following generation has also been tested in this species. However, even if larval development was improved when rearing offspring on a matching high-quality host plant, no trans-generational behavioral effects on host choice in young larvae and on female oviposition preferences were found [46]. In another moth species, the European grapevine moth *L. botrana*, larval odor experience during feeding on different grapevine cultivars led to female oviposition preference for the experienced cultivar, whereas early adult experience during hatching did not induce any preference [47]. Associative learning of aversive signals, i.e., associating an odor with an electric shock, also persisted trough metamorphosis in the tortricid moth *G. molesta* [48]. Another example for odor experience influencing host plant attractivity has been shown in the blue willow beetle, *Phratora vulgatissima,* a chrysomelid specialist of willows. This species has an innate preference for willow species with a low content of phenolic glucosides. Attractivity of volatiles emitted by different willow species can, however, be modified by experience and thus influence long-range orientation towards stands of willow trees. At close range, however, female preference to feed and oviposit on willow species containing less phenolic glycosides was not modified by previous experience [49].

Neural mechanisms underlying memory formation after experience with plant volatiles have not been specifically investigated in herbivorous insects to our knowledge. However, whereas adult experience with olfactory cues in general leads to synaptic strengthening in neural circuits and molecular as well as anatomical modifications in the brain, the question of how larval memory persists into adult life in holometabolous insects, which undergo remodeling of their central nervous system during pupation, is still not understood [50]. In a study on the beetle *Tenebrio obscurus*, however, increased RNA synthesis within the mushroom bodies has been described as a substrate for memory persisting through metamorphosis [51].

### 5.2. Learning of Odors through Association with Gustatory Signals

Odors experienced during feeding both by larvae and adults can influence subsequent orientation behavior. This is the case in the citrus psyllid *D. citri*, where adults learn to associate certain volatile compounds, such as limonene and benzaldehyde (and also colors), with a sucrose feeding reward and will more likely probe a surface in presence of the conditioned stimulus. Psyllids were able to associate blue color, which is not attractive to unconditioned psyllids, with a food reward when it was scented with citrus odors [52]. In the same species, experience with another compound, eugenol, leads to an aversive response in subsequent behavioral tests [52].

Feeding experience also influences odor response in locusts: starved individuals learn to associate odors (and also visual cues) with high reward food sources in a Y-maze experiment [53]. Odor learning can also be used to avoid ingesting toxic food in this species [54]. Food quality can play a role in learning effects. This is the case for example in *S. littoralis* where larval exposure to host plant volatiles influences larval host choice and oviposition preferences in adult females. A poor diet proposed with the host plant volatiles leads to a decrease in attraction of the associated plant, whereas association with a high-quality diet leads to increased attractivity of the corresponding plant [55]. However, in the case of the black vine weevil, *Otiorhynchus sulcatus*, adult feeding experience induced a preference for the experienced host plant, even if it was less suited for nutrition and survival due to higher phenolic compound content [56].

The neural mechanisms underlying the interaction between learned olfactory cues and a gustatory reward have been extensively studied in model insects such as the fruitfly and the honeybee [57,58], but these are beyond the scope of the present review.

### 5.3. Plasticity in Contact Compound Responses through Gustatory Experience

Feeding experience with different types of taste compounds has been shown to influence subsequent plant acceptation and feeding behavior, and these changes are often correlated with modifications in taste receptor responses and gustatory receptor expression (Figure 1). Larvae of different Lepidoptera change their food acceptance behavior after experience with host plants containing antifeedants. In larvae of *Pieris rapae*, rearing on a plant containing antifeedants leads to a lack of deterrence in response to different deterrent compounds (strychnine, naringin, and chlorogenic acid) present if larvae are reared on cabbage. At the same time, taste neurons on the maxillary palps reduce their sensitivity to these compounds, thus desensitization of neurons could be at the origin of behavioral desensitization [59]. Similar changes in the sensitivity of gustatory receptors as a function of feeding history were shown in larvae of *Papilio machaon* and *P. hospiton*, two highly specialized butterfly species. Sensitivity to individual compounds in both acceptance- and deterrence-mediating gustatory receptor neurons differed as a function of the plant the larvae were feeding on, indicating a sensitivity shift upon experience. Interestingly, response profiles of gustatory neurons in the two related species converged when larvae fed on the same host plant, whereas they differed if they were fed on different plants [60]. Larvae of the generalist moth species *M. sexta* narrow their host preference upon experience with certain host plants, such as solanaceous plants containing the steroidal glycoside indioside D, and become more specialist feeders. This change is correlated with enhanced responses of gustatory receptor neurons to indioside D, thus representing another example of experience-mediated changes in gustatory receptor neuron sensitivity leading to a change in larval feeding preferences [61]. Experience-dependent adaptation of gustatory neuron sensitivity also occurs for feeding stimulants. In larvae of *Helicoverpa armigera*, rearing on high-concentration sucrose diet leads to a transgenerational desensitization of sucrose-sensitive neurons. However, introducing larvae of any subsequent generation to a low-sucrose diet establishes re-sensitization very rapidly within the same generation [62]. In addition to physiological effects of experience on gustatory receptor neurons in various insects, anatomical modifications at the peripheral level have been found in locusts. Experience with food flavors added to the diet at the larval stage causes an increase in the number of contact chemosensory sensilla on the maxillary palps in later developmental stages and leads to a shorter feeding latency and longer feeding duration on plain food [63]. Another example of experience-dependent changes in response to gustatory signals is the cockroach adaptation to experience with toxic baits using sucrose as a phagostimulant. The occurrence of sucrose-averse cockroaches seems to be mediated through mutations in gustatory bitter receptors, which respond to glucose, and this stimulation suppresses activation of sucrose (sweet) receptors (activated in both wild type and mutated cockroaches) [64].

## 6. Influence of Physiological State on Host Plant Responses

Behavioral responses to host-plants or food sources depend on a variety of parameters, including feeding state, mating state, age, period within reproductive cycle, sex, and morph for polymorphic species (Table 1). The changes in these state-dependent responses very often have similar underlying mechanisms reaching from state-dependent variations in gene-expression at the peripheral and central nervous level, over receptor neuron and central neuron function, as well as neuromodulator action, up to anatomical changes in peripheral and central chemosensory structures. A common mechanism underlying various physiological state-dependent changes was found in the oriental fruitfly, *Bactrocera dorsalis.* In this species, mating state, feeding state, and the circadian rhythm caused changes in the expression of antennal chemosensory genes, which might ultimately lead to behavioral changes [65] (Figure 1).

### 6.1. Nutritional State and Symbiotic Bacteria can Modify Responses to Food- and Host-Related Volatiles

Whereas effects of the nutritional state on host-seeking behavior and underlying mechanisms have been broadly investigated in blood-feeding insects ([66] and references therein) and parasitoids [67], very little is known about such effects in herbivorous insects. However, decisions between the search for food sources or oviposition sites depend on previous food intake history in female herbivores as well (Table 1). Nutritional state-dependent variations in behavior and its underlying neural mechanisms have been studied extensively in *D. melanogaster* (for review see [68]). Recently the neural mechanisms underlying the choice between feeding and sexual behaviors have also been investigated. The nutritional state of fruit flies activates tyramine signaling as a mediator of satiety, which causes inhibition of feeding behavior and favors responses to signals involved in mating, whereas starved flies always orient towards food-related signals [69]. The feeding state can influence foraging behavior directly but is also dependent on gut bacteria. In *B. dorsalis*, foraging in protein-deprived flies is more goal-oriented in aposymbiotic flies than in flies with intact intestinal bacteria [70]. Similarly, olfactory preferences for cues emitted by food sources and associated microbes depends on gut microbiota in *D. melanogaster* [71]. However, facultative symbionts in several highly specialized biotypes of the pea aphid, *Acyrthosiphon pisum*, seem not to influence host plant selection [72]. To our knowledge, the influence of the wide-spread endosymbiont *Wolbachia* on host plant recognition in insects has not been investigated so far. This would be interesting to study, because *Wolbachia* has been shown to influence various behaviors in insects, such as locomotor activity, sleep, learning, feeding, and aggression [73,74,75].

### 6.2. Pronounced Effects of Age and Mating State on Female Responses to Host-Related Volatiles

In most insects, attraction to host- or food-related volatiles depends on mating state and age (Table 1). In female herbivores, egg maturation and egg production rate can also modulate attraction to host plant volatiles, often with an initial increase during early adult life and, in certain long-living insects, a decrease with advanced age. For example, females of the Queensland fruit fly, *Bactrocera tryoni*, were highly attracted to guava fruit juice in early adult life, but attraction declined with advanced adult age. However, this decline was correlated with a decline in locomotor activity (potentially due to decreasing oviposition rates) and once insects started to move, they still oriented towards the fruit odor [76]. In another study, mated females of the same species were more strongly attracted by fruit stimuli than virgin females and mated females spent more time on a fruit when reaching it [77]. In different psyllids, host choice is more selective in mated individuals. In the citrus psyllid *D. citri*, host plant volatiles attracted only mated males and females [37]. In the carrot psyllid, *Trioza apicalis,* host plant volatiles were overriding light intensity cues in mated females when competing visual and olfactory host cues were tested, whereas virgin females chose volatiles from a non-host plant in strong light conditions over a host plant under weak light [78].

Season-dependent changes in host-searching females have been found in the long-lived moth *Caloptilia fraxinella*, which undergoes an adult diapause during winter. Females showed weaker behavioral and electroantennogram (EAG) responses to host plant cues in the foliage of ash trees in winter than in summer. In addition, in summer, fed and reproductively active females exhibited stronger responses to the same cues than unfed or virgin females [79]. Furthermore, in *D. suzukii*, summer and winter morphs occur in northern temperate zones. Their behavioral and EAG responses to host fruit volatiles are significantly reduced in the winter morph concerning both attractive and repellent cues [80]. An age effect was also found in female melon flies, *Zeugodacus cucurbitae*, in attraction to volatiles emitted by preferred host plants with an increase in attraction during their lifetime, with the strongest attraction occurring when they reached sexual maturity at approximately four weeks of age. Less preferred hosts, however, did not vary in their attractivity as a function of female age [81].

### 6.3. Sex-Dependent Responses to Host Plant Volatiles

Various differences in host plant volatile responses have been observed between male and female insects depending on their biology (Table 1). In herbivorous insects where adult males do not feed, host plant volatiles alone often do not attract males as in the grapevine moth *L. botrana* [82]. In some cases, such as *Trichoplusia ni,* host plant volatiles can attract males on their own, however [83]. More commonly, host plant odors act on males in synergism with pheromones as an additional cue to localize females (reviewed in [84]). In several moth species, such as *Cydia pomonella* and *G. molesta*, host plant volatiles synergize responses to female-emitted sex pheromones [85,86]. In male *Helicoverpa zea*, host plant volatiles were even found to synergize responses to the sex pheromone in pheromone-specific olfactory receptor neurons [87]. Cases of synergistic attraction of host plant volatiles with the female produced sex pheromone were also reported in the scarab beetle *Holotrichia parallela* [88]. In the oriental fruit fly *D. suzukii*, the effect of blends versus individual compounds has been shown to vary between the sexes. Certain individual fruit volatiles and specific blends of compounds are attractive to females, whereas single compounds do not attract males. However, a blend of two specific volatiles, beta-cyclocitral and isoamyl acetate, attracts males specifically but not females [89]. In aphids, such as *Sitobion avenae* and *S. fragariae,* winged females under autumn conditions are strongly attracted by their respective host plant odors, whereas males in these two species do not respond to host plant odors, but only to the female-emitted sex pheromone. In another aphid species, *R. padi*, however, even males respond to the odor of their primary host (reviewed in [90]).

### 6.4. Morph-Dependent Responses to Host Plant-Related Volatiles

Different morphs of the same insect (with identical genomes) can also vary in behavioral and physiological responses to host plant volatiles (Table 1). In certain aphid species, high population densities lead to the formation of winged morphs, to facilitate dispersal [91]. In addition to the increased number of olfactory organs on the antennae, evidenced in winged morphs of different species [92,93], stronger preferences to host plant odors have also been identified, for example in *M. persicae* [30]. In the same aphid species, responses to the host plant potato were shown to be influenced by volatile emissions caused by a companion plant (onion) but the level of response was morph dependent: winged aphids were repelled by high doses of onion-induced terpenoids (nerolidiol, TMTT) emitted from potato, whereas wingless morphs were only repelled by low doses of the same compounds [30]. To our knowledge no studies comparing responses to contact cues have been conducted between morphs so far.

Similar to aphids, locusts also occur in two morphs depending on population density. They switch from solitary to gregarious forms, which differ considerably in their sensory equipment and responses to host plant volatiles. Solitary locusts possess higher numbers of antennal chemoreceptors [94] and different expression of antennal odorant binding proteins compared to gregarious locusts [95]. Larger numbers of receptor neuron axons converge within the primary olfactory centers in solitary locusts, leading to larger relative sizes of the antennal lobes compared to gregarious locusts [96,97]. Central olfactory neurons respond more frequently to plant volatiles, corresponding to a higher sensitivity and localization over larger distances of host plants, as well as more selectivity in host plant choice [98,99] (Figure 1). Interestingly, learning capacities are also rather different between solitary and gregarious locusts. Whereas appetitive learning is similar in both phases, aversive learning of odors paired with exposure to toxic plant compounds occurs more rapidly in solitary locusts, thus eliciting immediate avoidance of plants, whereas gregarious locusts need more time to associate these signals, probably because the association is in this case linked to post-ingestive learning pathways. Faster learning in solitary locusts might be due to higher peripheral sensitivity to toxic contact stimuli, or modulatory mechanisms in the brain [99].

## 7. Chemosensory Plasticity Leading to Host Plant Adaptation

At the evolutionary level, diversification in the use of host plants, possible through the plasticity of host choice mechanisms, might also lead to plasticity-driven speciation, i.e., host plant adaptation. *Drosophila* species are interesting models to study adaptational plasticity in host plant recognition. *D. suzukii* for example is a species which moved its egg-laying preferences from rotten to fresh fruit. Due to its behavioral and physiological plasticity, *D. suzukii* is a highly adaptable species, and has therefore become one of the most important invasive agricultural pest insects in large parts of the world [100]. One of the plasticity traits is the change in receptor genes responding to the host volatile, isoamyl acetate, commonly emitted by fresh fruit, which was shown to be highly attractive to *D. suzukii.* Interestingly, one of the receptor genes activated by isoamyl acetate, OR67a, is represented by five duplicated copies, potentially providing a molecular mechanism underlying the described host shift [101]. In another *Drosophila* species, *D. erecta*, specialized on screw pine fruits, a similar type of adaptation was found. Here, one type of olfactory receptor neurons (ORNs) increases sensitivity to a characteristic host volatile, 3-methyl-2-butenyl acetate. The number of these specific ORNs is higher on the antennae, and the glomeruli receiving their input within the antennal lobe are enlarged compared to sibling species [102] similar to *D. sechellia*, with the same type of sensory adaptations to a special host, morinda fruit, and one of its major volatiles, methyl hexanoate [103] (Figure 1).

The ability to feed on different host plants might also have paved the way for specialization and eventually speciation in nymphalid butterflies. Here, a broad range of host plants in ancestral species might have led to adaptation to feed on suboptimal host plants at a long-term scale to evolutionary innovation and new species with a change in host range [104]. The molecular mechanisms underlying evolutionary host adaptation at the antennal level have recently been investigated in different Hemiptera. In the white-backed planthopper *Sogatella furcifera*, a specific OR gene clade was identified, which diverged strongly from other hemipteran ORs. This may indicate that these novel ORs might be involved in the adaptation of their host plant (rice) [105]. As a first step in the process of specialization, host plant transfer in the alfalfa bug *Adelphocoris lineolatus* led to changes in the antennal transcriptome. The expression of different chemosensory proteins varied as a function of the host plant [106]. This indicates that sensory plasticity at the antennal level contributes to the adaptation to a specific host plant in this mirid bug.

In addition to plasticity in response to individual volatiles contributing to the evolutionary adaptation of species with broad host ranges, plasticity in the integration of different cues (either synergistic or contradicting) also contributes to the strong adaptability to varying environmental conditions seen in bark beetles [26]. In the European spruce bark beetle, *Ips typographus*, signals originating from non-host trees interact with host tree volatiles and can change their long-range orientation behavior. Small modifications in the way different signals are integrated can thus lead to changes in host-plant preference and adaptation to a changing environment [26].

Contact stimuli, such as those detected by contact chemoreceptors on the ovipositor during oviposition, can also be involved in host plant adaptation processes. Invasive plants with similar attractive cues as native host plants but containing toxic substances, such as glucosinolates, in larger amounts than native host plants might create an evolutionary trap by killing offspring after females choose these invasive plants for oviposition. However, in the butterfly *Pieris macdunnoughii,* frequent encounters with the native (less toxic) host plant increased preference over the invasive (toxic) plant, thus avoiding the evolutionary trap [107].

## 8. Discussion and Perspectives towards Application

Efforts to replace classical pest management strategies using synthetic insecticides with alternative strategies have been made since many years. However, recent developments in the banning of hazardous insecticides require more research on alternative pest management strategies, and chemical ecology provides promising avenues for behavior-based control strategies using plant compounds. Such methods comprise the use of repulsive and attractive compounds in dispensers and/or companion plants and possibly their essential oils, either in combination with confined insecticides in an attract and kill approach, or the combination of repulsive and attractive elements in push–pull systems. Other approaches include masking of host plant compounds to disrupt recognition, selection of crop varieties emitting less attractive volatiles, and manipulation of volatile emission in plants by plant resistance induction through plant defense stimulators [108,109].

Chemical signals, alone or in concert with other sensory cues, are essential for host plant recognition and choice, but determining the identity of attractive or repulsive compounds is far from being sufficient to develop efficient and reliable crop protection methods based on these signals. First, collection of potentially active volatiles needs to be performed at the correct physiological state of the host or non-host plant. Second, the complexity of the signal and interactions between various host plant cues need to be considered. Indeed, as shown in the present review, there are multiple interaction levels between various host plant cues and their environment (Figure 2). The knowledge on chemical cue interactions is important for potential applications, because the use of attractive or repulsive plants is costly both in time and money, and therefore, as a first step, identification of individual volatile compounds, which can potentially be produced at a reasonable cost and applied relatively easily is often attempted. Nevertheless, it is difficult to reach high efficiency with this approach. One example is the use of individual volatile compounds present in alternative host plants for the spotted lanternfly, *Lycorma delicatula*. One individual compound, sulcatone, proved highly attractive, but blends of different identified compounds further improved attraction with a potential to develop lures for field testing [110]. At close range, attempts are made to use treatments on plant surfaces with aversive compounds. Treatments of citrus seedlings with alkaloid plant extracts and certain compounds identified within the extracts are able to reduce attraction to the host plants for the Asian citrus psyllid, *D. citri* and thus represent potential for reducing the colonization of citrus plants. Again, synergistic effects of several compounds were found [111]. Even though volatile compounds may lead to important decisions in herbivores while approaching potential host plants, one should keep in mind that the final acceptance still depends on contact chemical cues (e.g., [17] and references therein).

Furthermore, as shown in the present review, insect behavior is highly plastic (Figure 2) and therefore the different forms of plasticity need to be taken into account when developing pest control methods involving the chemical senses. Insects with the same genome might not respond in the same way to chemical stimuli as a function of various factors, such as experience, physiological state, and season of the year. Powerful attractants or repellents might change their efficiency through adaptation and learning processes, as shown for the above-described case of a switch in the effect of sucrose from a phagostimulant to an aversive substance in cockroaches experienced with toxic baits [64]. Attractive, repellent, or stimulating compounds might not have the same effects on mature and immature insects. This could be important in pests such as the cabbage-stem flea beetle, *Psylliodes chrysocephala*, where immature females represent the most important plant-colonizing state while the physiological state of insects tested in behavioral experiments is often not considered (e.g., [112], but see observed differences in volatile responses between spring and early fall individuals of the crucifer flea beetle *Phyllotreta cruciferae* [113]). Furthermore, evaluating the effect of chemical signals on both males and females may be important, as both sexes may not have the same response and, depending on the case, either one or both sexes feed on the plant and damage the crop. Whereas host plant attraction of females is ubiquitous in most investigated insects, male attraction in species where adults do not feed is more rare and often only efficient in combination with sex pheromones but can improve reproductive success [84]. A differential evaluation of behavioral responses according to experience, physiological state, and/or sex could help in selecting compounds, combinations of compounds, doses or ratios affecting appropriate targets, and fine-tune behavioral manipulation of pests to protect crops but also maintain efficiency. Indeed, even when repellent volatiles from non-host plants are identified, experience-dependent plasticity may compromise long-term use of such treatments. This has been shown for example in the diamondback moth, *Plutella xylostella*, where experience with an extract from a non-host plant, *Chrysanthemum morifolium*, applied on a host plant rapidly reduces the repellent effect and even leads to attraction to host plants treated with the non-host extract [114]. Other important points to consider are the nutritional state of an insect, and seasonal and age changes in responses to chemical cues as described in Section 6.1 and Section 6.2. Treatments or trapping should be considered at the moment when insects are most susceptible to the used cues.

When looking into mechanisms, plasticity in plant volatile detection and processing can lead to changing behavioral preferences for host plants (Figure 1). While this is supposed to be an adaptational process during evolution leading to an optimal host plant choice, changes in the algorithms of VOC processing in the insect brain can occasionally lead to errors in host plant identification [115]. Gaining knowledge on mechanisms of VOC processing algorithms in insects could allow manipulation of female attraction to direct them to unsuitable plants and reduce pest populations.

Even though we did not include abiotic effects in the present review, in addition to sensory cue interactions and the different forms of plasticity, abiotic factors (such as CO_2_, nitrogen, and ozone) can influence the attraction of insects to their host plants, and their influence should be considered in future behavior-based pest control measures. These abiotic factors are of increasing importance due to global changes in the environment. Different mechanisms can be involved, such as changes in plant metabolisms, leading to changes in volatile emissions, transformation of compounds by abiotic factors in the environment, and potentially modification of sensitivity in the receiver organism [116]. In insects, modifications to olfactory receptor neuron signal transmissions by abiotic factors, such as air pollution, have not yet been found, but should definitely be studied in the future. In plants, however, cell membrane modifications by pollutants have been evidenced, leading to changes in plant internal signaling [116,117].

This review shows that mechanisms of insect host plant localization and recognition are highly plastic. These mechanisms are an excellent target for the development of alternative pest control methods. Some of these are already widely used but have a significant potential to be further improved and extended to a large range of pests. In particular, their use may have been oversimplified, while we show here that plasticity both among plant cues or in behavioral responses of insects may strongly influence their efficiency and durability of use. Based on the rich literature reviewed in the present paper, we hope to see consideration of plasticity in future research on next-generation, sustainable, pest insect control based on behavioral manipulation.

## Figures and Tables

**Figure 1 biology-11-01842-f001:**
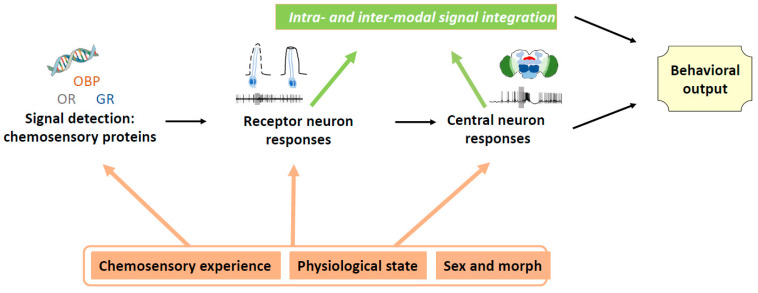
Mechanisms involved in chemosensory plasticity at different integration levels, leading to modulation of behavioral output. GR gustatory receptor, OBP odorant binding protein, OR odorant receptor.

**Figure 2 biology-11-01842-f002:**
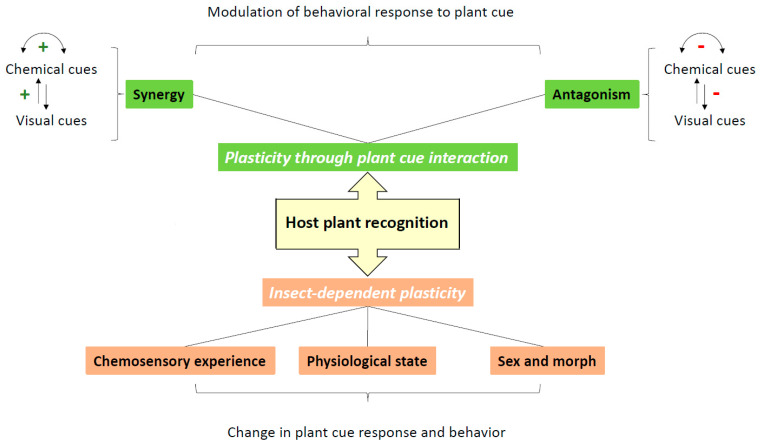
Overview over the different origins of plasticity in host plant recognition.

## Data Availability

Not applicable.

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
