# Peer review of "Plasticity in Chemical Host Plant Recognition in Herbivorous Insects and Its Implication for Pest Control"

_biology, 2022, doi:10.3390/biology11121842_

Round 1

Reviewer 1 Report

This review presents in a comprehensive, clear, and coherent way the research carried out around the phenomenon of phenotypic plasticity in the phytophagous-host interaction, and as a very important element and little analyzed to date, the implications for pest control, as well as research prospects.

Some minor fixes are included in the attachment.

Below are some references that authors should consider including

Thöming, G., Larsson, M. C., Hansson, B. S., & Anderson, P. (2013). Comparison of plant preference hierarchies of male and female moths and the impact of larval rearing hosts. Ecology, 94(8), 1744–1752. https://doi.org/10.1890/12-0907.1

TY  - JOUR

AU  - Anton, Sylvia

AU  - Rössler, Wolfgang

PY  - 2021

DA  - 2021/01/01

TI  - Plasticity and modulation of olfactory circuits in insects

JO  - Cell and Tissue Research

SP  - 149

EP  - 164

VL  - 383

IS  - 1

AB  - Olfactory circuits change structurally and physiologically during development and adult life. This allows insects to respond to olfactory cues in an appropriate and adaptive way according to their physiological and behavioral state, and to adapt to their specific abiotic and biotic natural environment. We highlight here findings on olfactory plasticity and modulation in various model and non-model insects with an emphasis on moths and social Hymenoptera. Different categories of plasticity occur in the olfactory systems of insects. One type relates to the reproductive or feeding state, as well as to adult age. Another type of plasticity is context-dependent and includes influences of the immediate sensory and abiotic environment, but also environmental conditions during postembryonic development, periods of adult behavioral maturation, and short- and long-term sensory experience. Finally, plasticity in olfactory circuits is linked to associative learning and memory formation. The vast majority of the available literature summarized here deals with plasticity in primary and secondary olfactory brain centers, but also peripheral modulation is treated. The described molecular, physiological, and structural neuronal changes occur under the influence of neuromodulators such as biogenic amines, neuropeptides, and hormones, but the mechanisms through which they act are only beginning to be analyzed.

SN  - 1432-0878

UR  - https://doi.org/10.1007/s00441-020-03329-z

DO  - 10.1007/s00441-020-03329-z

ID  - Anton2021

ER  - 

Sant’Ana, J., Tognon, R., Pires, P., & Gregório, P. (2021). Associative learning and memory through metamorphosis in Grapholita molesta (Busck) (Lepidoptera: Tortricidae). The Canadian Entomologist, 153(6), 666-671. doi:10.4039/tce.2021.39

Fred Punzo, Richard J Malatesta,

Brain RNA synthesis and the retention of learning through metamorphosis in Tenebrio obscurus (insecta: coleoptera),

Comparative Biochemistry and Physiology Part A: Physiology,

Volume 91, Issue 4,

1988,

Pages 675-678,

ISSN 0300-9629,

https://doi.org/10.1016/0300-9629(88)90947-4.

(https://www.sciencedirect.com/science/article/pii/0300962988909474)

Ninkovic, V., Dahlin, I., Vucetic, A., Petrovic-Obradovic, O., Glinwood, R., & Webster, B. (2013). Volatile Exchange between Undamaged Plants - a New Mechanism Affecting Insect Orientation in Intercropping. PLOS ONE, 8(7), e69431. https://doi.org/10.1371/journal.pone.0069431

Author Response

We thank the reviewer for the positive comments and suggestions. We included now the suggested references in the manuscript at the appropriate sections (in the introduction page 2, part 3. Page 8, and part 5.1. page 9) and corrected the indicated errors in the text.

Reviewer 2 Report

This is a very good comprehensive work which expresses recent and environment safe trend of plant protection strategies. Many of the examples have been shown as it has been presented in earlier review work e.g. Marcia O. Mello and Marcio C. Silva-Filho (2002). Plant-insect interactions: an evolutionary arms race between two distinct defense mechanisms. Brazilian Journal of Plant Physiology, vol 14 no.2. More case oriented examples may strengthen the manuscript to satisfy the need of applied plant protection. 

Author Response

We thank the reviewer for the positive evaluation.

We have now included the suggested reference in the introduction page 2 to state more clearly which aspects we cover in the present manuscript.

Concerning case-oriented examples, we are not sure what the reviewer proposes. We cite many specific cases throughout the manuscript and base our discussion on these, to highlight that plasticity is important to be taken into account in the application of chemical ecology to plant protection.

Reviewer 3 Report

A very interesting review article and excellent work to put it together. I only have the following minor comments for authors' consideration.

Abstract:

Line33-34, not clear “plasticity in plant choice behavior”, does it mean insect choice or plant choice?

Line 36-37, probably need to add herbivores and plant interactions

Introduction:

Line 57-58, host plant choice can also be selected by males as well. How larvae approach host plants?

Line 89-90, VOCs are emitted that can be not naturally (e.g. plant defensive compounds) may also act long-range stimuli.

Authors discussed the long-range stimuli and short distance attractants, what is the definition for “Long” and “short”. Aphid can fly relatively long towards their pheromone-based traps in the field.

Line 285-288, flower visiting moths, such as M. sexta, are nocturnal, which obviously vision won’t play key roles in host plant location.

Line 340, “(but also colors)” could authors provide some details on this interesting topic, and how the differences between olfactory cue+color vs contact sense+colors?

Line 473-475, the weak EAG responses could also be related to lower temperatures during winter, or less volatile cues from the plants, not sure if ages would play some roles here dominantly, or not.

Line 610, suggest rephrasing the work “important” to “hazardous” pesticides

Line 615, may add some discussion on agitating pest behavior via spatial stimulants with insecticides

Author Response

We thank the reviewer for the positive comments and suggestions. We considered all comments as follows and made the corrections in the text.

Line33-34, not clear “plasticity in plant choice behavior”, does it mean insect choice or plant choice?

Re: We changed to: insect plant choice behaviour (we mean insects choosing plants)

Line 36-37, probably need to add herbivores and plant interactions

Re: We reformulated the sentence to “…combined effects between long-and close-range chemical cues in herbivore-plant interactions….

Introduction:

Line 57-58, host plant choice can also be selected by males as well. How larvae approach host plants?

Re: We changed to: line 53: “…issue for all male and female insects” and line 57: “…by adult insects, but also larvae hatched on the ground can approach host plants, or larvae can switch host plants…

Line 89-90, VOCs are emitted that can be not naturally (e.g. plant defensive compounds) may also act long-range stimuli.

Re: We omitted “naturally” (even though we believe that plant defensive compounds are still compounds which are naturally emitted as compared to artificially applied compounds).

Authors discussed the long-range stimuli and short distance attractants, what is the definition for “Long” and “short”. Aphid can fly relatively long towards their pheromone-based traps in the field.

Re: long-and short-range is a relative measure, which depends on insect size and mobility, as such it is difficult to provide a precise definition. Concerning aphid movements over long distances: they are most likely passively transported with air movements, as stated in the text and this can happen over large distances. However, decisions, for example a response to pheromone traps, are to our knowledge only taken at relatively short distances (and wind-transported aphids drop when they sense the odor).

Line 285-288, flower visiting moths, such as M. sexta, are nocturnal, which obviously vision won’t play key roles in host plant location.

Re: Even though M. sexta is night-active, visual cues are involved at least in close range orientation, as explained in the cited references.

Line 340, “(but also colors)” could authors provide some details on this interesting topic, and how the differences between olfactory cue+color vs contact sense+colors?

Re: We have now added a sentence in the text to explain learning of color cues in combination with citrus odors.

Line 473-475, the weak EAG responses could also be related to lower temperatures during winter, or less volatile cues from the plants, not sure if ages would play some roles here dominantly, or not.

Re: We agree with your analysis. However experiments were performed in controlled conditions with identic temperatures and the same type of plant cue (ash seedlings). We took, however the term “age-dependent” our in the first sentence of this paragraph.

Line 610, suggest rephrasing the work “important” to “hazardous” pesticides

Re: We changed the wording accordingly

Line 615, may add some discussion on agitating pest behavior via spatial stimulants with insecticides

Re: We are very sorry, but we do not understand what is meant with this comment. Do you mean that we should discuss that insect locomotor behaviour can be increased (and decreased), when exposed to sublethal doses of insecticides? However in the sentence we mention only the use of confined insecticides in an attract-and-kill approach, where sublethal effects would most likely not occur.